# Time-Domain Sound Field Reproduction with Pressure and Particle Velocity Jointly Controlled

**Xuanqi Hu, Jiale Wang, Wen Zhang ***  **and Lijun Zhang**

Center of Intelligent Acoustics and Immersive Communications, School of Marine Science and Technology, Northwestern Polytechnical University, Xi'an 710072, China; 2019200558@mail.nwpu.edu.cn (X.H.); wang.jl@mail.nwpu.edu.cn (J.W.); zhanglj7385@nwpu.edu.cn (L.Z.)
* Correspondence: wen.zhang@nwpu.edu.cn

**Abstract:** Particle velocity has been introduced to improve the performance of spatial sound field reproduction systems with an irregular loudspeaker array setup. However, existing systems have only been developed in the frequency domain. In this work, we propose a time-domain sound field reproduction method with both sound pressure and particle velocity components jointly controlled. To solve the computational complexity problem associated with the multi-channel setup and the long-length filter design, we adopt the general eigenvalue decomposition-based approach and the conjugate gradient method. The performance of the proposed method is evaluated through numerical simulations with both a regular loudspeaker array layout and an irregular loudspeaker array layout in a room environment.

**Keywords:** sound field reproduction; pressure matching; particle velocity; time domain

## 1. Introduction

Spatial sound field reproduction aims at faithfully reproducing the sound field within an extended region of space so that listeners inside the region could experience the replication of the original sound field as realistically as possible. Such a system normally uses multiple loudspeakers as secondary sources to control sound radiation and nowadays has been widely used in various audio applications such as in theaters, cinemas, concerts, home entertainment systems, etc.

The exploration of spatial sound field reproduction has never stopped, up to now, many approaches have been developed for this technology, including wave field synthesis (WFS), Ambisonics, and least-squares (LS)-based multi-point control [1]. The WFS approach was first proposed by Berkhout, which is based on the Huygens–Fresnel integral to represent a sound field within a bounded region of the space by a continuous distribution of monopole and dipole secondary sources arranged on the boundary of that region [2,3]. For practical implementation, an array of equally spaced loudspeakers are used to approximate the continuous distribution of secondary sources. Reproduction artifacts due to the spatial discretization of the continuously-distributed secondary sources and the finite size of the array were investigated [4]. In WFS, a large number of closely spaced loudspeakers is necessary for accurate sound field reproduction.

Another well-known sound field reproduction technique, Ambisonics, was designed based on Huygens principle [5,6]. The system adopts the zero and first order spherical harmonic decomposition of the original sound field into four channels, and from a linear combination of these four channels to derive the loudspeaker driving signals. This low-order system is optimum only at low frequencies and when the listener stays at the central sweet spot. Later, Higher-Order Ambisonics (HOA) based on the higher-order spherical harmonic decomposition of a sound field was developed for high reproduction frequencies and large reproduction regions [7–9]. A typical Ambisonic or HOA system uses a circular or spherical loudspeaker array geometry. In addition, the spherical harmonic expansion

order increases with the reproduction frequency and radius of the reproduction region, thus in most time HOA also requires densely distributed loudspeakers to match all the spherical harmonics to the given order to avoid spatial aliasing [10].

For arbitrarily placed loudspeakers, a simple approach in sound reproduction is the least-squares (LS) based multi-point control, which uses multiple microphones as matching points to derive the least-square solution as the loudspeaker weights [11]. This approach is a typical inverse problem, and Tikhonov regularization is commonly adopted to obtain the loudspeaker weights with limited energy and to improve the system robustness. With this method, the placement of both loudspeakers and matching microphones greatly affects control accuracy and filter stability [12]. In addition, the acoustic characteristics of loudspeakers affect the reproduction results, and its modeling though measurements has been discussed in previous work [13].

While most existing work focuses on controlling sound pressure only, some recent work started to investigate controlling the particle velocity in sound field reproduction [14]. A joint control of sound pressure and particle velocity has also been proposed for single-zone [15] and multi-zone sound field reproduction [16]. A general finding is that particle velocity assisted sound field reproduction is especially suitable to a non-uniformly spaced loudspeaker array with reduced number of loudspeakers and control points required. The extension to intensity based sound field reproduction has also been investigated [17,18].

So far, the particle velocity assisted sound field reproduction system has only been developed in the frequency domain. In this work, we propose a time-domain sound field reproduction algorithm with both sound pressure and particle velocity jointly controlled. As demonstrated in many works, time-domain processing in spatial sound recording and reproduction is suited for real-time applications [19,20]; however, it is also computationally expensive as long-tap room impulse response (RIR) filters are usually involved for sound field reproduction inside reverberant rooms. We adopt the eigenvalue decomposition (EVD)-based approach and the conjugate gradient (CG) method [21,22] in this work to reduce the computational complexity.

The paper is organized as follows. Frequency-domain velocity assisted sound field reproduction is reviewed in Section 2. In Section 3, the proposed time-domain sound field reproduction with joint control of sound pressure and particle velocity, and implementation details, are introduced. In Section 4, the effectiveness of the proposed method is evaluated through numerical simulations in a room environment of different reverbration times. Finally, Section 5 concludes this paper.

Notations: italic letters denote scalars, lower case boldface letters denote vectors, and upper case boldface letters denote matrices.

## 2. Review: Frequency-Domain Velocity-Assisted Sound Field Reproduction

As a starting point, we briefly review the concept of frequency-domain velocity assisted sound field reproduction. At an arbitrary observation position $\mathbf{x}$, the particle velocity $\mathbf{v}(\mathbf{x}, \omega)$ and the complex-valued sound pressure $p(\mathbf{x}, \omega)$ with time-dependency $e^{i\omega t}$ have a relationship established by Euler's equation,

$$-\nabla p(\mathbf{x}, \omega) = i\omega\rho\mathbf{v}(\mathbf{x}, \omega), \tag{1}$$

where $i$ is the imaginary unit, $\omega = 2\pi f$ denotes the angular frequency, $\rho$ is the density of the propagation medium, and $\nabla$ represents the gradient operation along the direction of the particle velocity vector. The components of the particle velocity vector can be defined either in the Cartesian coordinate, i.e., $\mathbf{v} \equiv \{\mathbf{v}_x, \mathbf{v}_y, \mathbf{v}_z\}$, or the polar coordinate, such as $\mathbf{v} \equiv \{\mathbf{v}_{\mathrm{rad}}, \mathbf{v}_\theta, \mathbf{v}_\phi\}$ along the radial direction, the elevation and azimuth angular direction, respectively.

In sound field reconstruction, we consider the reproduced sound generated by an array of $L$ loudspeakers located positioned at $\mathbf{y}_l$ with $\ell = 1, \ldots, L$ surrounding the listening area. We define the acoustic transfer function (ATF) for the sound pressure component from the $\ell$th loudspeaker to the control point $\mathbf{x}$ as $T_p(\mathbf{x}|\mathbf{y}_\ell, \omega)$. A special case is when the

loudspeakers are modeled as point sources, and by assuming free-field propagation, the ATF is represented by the Green's function, that is

$$T_p^{\text{free-field}}(\mathbf{x}|\mathbf{y}_\ell, \omega) = \frac{1}{4\pi} \frac{e^{ik\|\mathbf{y}_\ell - \mathbf{x}\|_2}}{\|\mathbf{y}_\ell - \mathbf{x}\|_2}, \tag{2}$$

where $k = \omega/c_0$ is the wave number, $c_0$ denotes the sound speed, and $\|\cdot\|_2$ denotes the $L2$-norm.

Then, the reproduced sound pressure at position $\mathbf{x}$ can be expressed as

$$\begin{aligned} p(\mathbf{x}, \omega) &= \sum_{\ell=1}^{L} T_p(\mathbf{x}|\mathbf{y}_\ell, \omega) w_\ell(\omega) S(\omega) \\ &= \mathbf{t}_p^T(\mathbf{x}, \omega) \mathbf{w}(\omega) S(\omega), \end{aligned} \tag{3}$$

where $\mathbf{t}_p(\mathbf{x}, \omega) = \left[ T_p(\mathbf{x}|\mathbf{y}_1, \omega), \dots, T_p(\mathbf{x}|\mathbf{y}_L, \omega) \right]^T$ is a column vector containing the ATFs for all the loudspeakers to the position $\mathbf{x}$, $\mathbf{w}(\omega) = [w_1(\omega), \dots, w_L(\omega)]^T$ is the vector consisting of the frequency-domain loudspeaker weights, and $S(\omega)$ is the source audio signal. $(\cdot)^T$ denotes the transpose operator.

Similarly, we can define the ATF for the particle velocity, i.e., $\mathbf{t}_v(\mathbf{x}|\mathbf{y}_\ell, \omega)$, which is a column vector of length 3 for each component of $\mathbf{v}$, and has the following representation for the reproduced particle velocity:

$$\begin{aligned} \mathbf{v}(\mathbf{x}, \omega) &= \sum_{\ell=1}^{L} \mathbf{t}_v(\mathbf{x}|\mathbf{y}_\ell, \omega) w_\ell(\omega) S(\omega) \\ &= \mathbf{T}_v^T(\mathbf{x}, \omega) \mathbf{w}(\omega) S(\omega), \end{aligned} \tag{4}$$

where $\mathbf{T}_v(\mathbf{x}, \omega) = [\mathbf{t}_v(\mathbf{x}|\mathbf{y}_1, \omega), \dots, \mathbf{t}(\mathbf{x}|\mathbf{y}_L, \omega)]^T$ is a matrix of size $L \times 3$.

Given that in sound field reproduction applications, we aim to reproduce the desired sound within a certain region of interest, by matching the sound pressure and particle velocity at multiple control points. That is, we have the matrix form representation of (3) and (4) as

$$\mathbf{p}(\omega) = \mathbf{T}_p(\omega) \mathbf{w}(\omega) S(\omega) \tag{5}$$

$$\mathbf{v}(\omega) = \mathbf{T}_v(\omega) \mathbf{w}(\omega) S(\omega), \tag{6}$$

where given $M$ control points, $\mathbf{x}_m$ and $m = 1, \dots, M$, $\mathbf{p}(\omega) = [p(\mathbf{x}_1, \omega), \dots, p(\mathbf{x}_M, \omega)]^T$ and $\mathbf{v}(\omega) = [\mathbf{v}^T(\mathbf{x}_1, \omega), \dots, \mathbf{v}^T(\mathbf{x}_M, \omega)]^T$ are column vectors of length $M$ and $3M$, respectively. The ATF matrix

$$\mathbf{T}_p(\omega) = [\mathbf{t}_p(\mathbf{x}_1, \omega), \dots, \mathbf{t}_p(\mathbf{x}_M, \omega)]^T$$

$$\mathbf{T}_v(\omega) = [\mathbf{T}_v(\mathbf{x}_1, \omega), \dots, \mathbf{T}_v(\mathbf{x}_M, \omega)]^T$$

is of size of $M \times L$ and $3M \times L$, respectively.

Based on Equations (5) and (6), and assuming the unit amplitude of the source audio signal, i.e., $S(\omega) = 1$, the cost function for a jointly controlling the reproduced sound pressure and particle velocity is formulated as follows:

$$\min_{\mathbf{w}(\omega)} \{ \tau(\omega) \| \mathbf{T}_p(\omega) \mathbf{w}(\omega) - \mathbf{p}_d(\omega) \|_2 + (1 - \tau(\omega)) \| \mathbf{T}_v(\omega) \mathbf{w}(\omega) - \mathbf{v}_d(\omega) \|_2 \} \tag{7}$$

where $\mathbf{p}_d(\omega)$ and $\mathbf{v}_d(\omega)$ represent the desired pressure and particle velocity, respectively. The control strategy is to minimize the reproduction error of both components, and $\tau(\omega) \in [0, 1]$ is the parameter to adjust the relative weights for matching of pressure and velocity. Equation (7) is known as the weighted least squares problem and can be solved using a Moore–Penrose pseudoinverse with Tikhonov regularization.

### 3. Proposed: Time-Domain Sound Field Reproduction with Joint Control of Sound Pressure and Particle Velocity

*3.1. System Formulation*

Now, we discuss the problem of sound field reproduction in the time domain. Assuming the room impulse responses (RIRs) are pre-calibrated through measurements, the reproduced sound pressure and particle velocity at the $m$th $(1 \leqslant m \leqslant M)$ control point $\mathbf{x}_m$, generated by $L$ loudspeakers located at $\mathbf{y}_1, \ldots, \mathbf{y}_L$, can be expressed as

$$p_n(\mathbf{x}_m) = \sum_{\ell=1}^{L} s_n * q_n^\ell * h_{p,n}(\mathbf{x}_m|\mathbf{y}_\ell) \tag{8}$$

$$\mathbf{v}_n(\mathbf{x}_m) = \sum_{\ell=1}^{L} s_n * q_n^\ell * \mathbf{h}_{v,n}(\mathbf{x}_m|\mathbf{y}_\ell), \tag{9}$$

where $*$ denotes the linear convolution operator and $n$ denotes the sampling index. $s_n$ denotes the input sound signal, $q_n^\ell$ denotes the control filter for the $\ell$th loudspeaker, $h_{p,n}(\mathbf{x}_m|\mathbf{y}_\ell)$ and $\mathbf{h}_{v,n}(\mathbf{x}_m|\mathbf{y}_\ell)$ denote the RIRs of pressure and velocity components, respectively, from the $\ell$-th loudspeaker to the $m$-th control point. Note that for particle velocity vector, we follow the convention to define three components along x, y and z axes, that is,

$$\mathbf{v}_n(\mathbf{x}_m) = [v_{n,x}(\mathbf{x}_m), v_{n,y}(\mathbf{x}_m), v_{n,z}(\mathbf{x}_m)]^T$$

and

$$\mathbf{h}_{v,n}(\mathbf{x}_m|\mathbf{y}_\ell) = [h_{v,n}^x(\mathbf{x}_m|\mathbf{y}_\ell), h_{v,n}^y(\mathbf{x}_m|\mathbf{y}_\ell), h_{v,n}^z(\mathbf{x}_m|\mathbf{y}_\ell)]^T.$$

Represent (8) and (9) in matrix form, we have

$$p_n(\mathbf{x}_m) = \sum_{\ell=1}^{L} \mathbf{q}_\ell^T \mathbf{H}_p(\mathbf{x}_m|\mathbf{y}_\ell)\mathbf{s}_n = \mathbf{q}^T \mathbf{H}_p(\mathbf{x}_m)\mathbf{s}_n \tag{10}$$

$$\mathbf{v}_{n,c}(\mathbf{x}_m) = \sum_{\ell=1}^{L} \mathbf{q}_\ell^T \mathbf{H}_{v,c}(\mathbf{x}_m|\mathbf{y}_\ell)\mathbf{s}_n = \mathbf{q}^T \mathbf{H}_{v,c}(\mathbf{x}_m)\mathbf{s}_n, c \in x, y, z, \tag{11}$$

where given the $K$-tap long RIR and $J$-tap long control filter,

$$\mathbf{s}_n = \left[s_n, s_{n-1}, \ldots, s_{n-(K+J-2)}\right]^T,$$

$$\mathbf{q}_\ell = \left[q_1^\ell, q_2^\ell, \ldots, q_J^\ell\right]^T,$$

and the RIR matrices $\mathbf{H}_p(\mathbf{x}_m|\mathbf{y}_\ell)$ and $\mathbf{H}_{v,c}(\mathbf{x}_m|\mathbf{y}_\ell)$ are Toeplitz matrices of size $J \times (K + J - 1)$. The first row vector and the first column vector of $\mathbf{H}_p(\mathbf{x}_m|\mathbf{y}_\ell)$ are defined as

$$\left[h_{p,1}(\mathbf{x}_m|\mathbf{y}_\ell), \ldots, h_{p,K}(\mathbf{x}_m|\mathbf{y}_\ell), \underbrace{0, \ldots, 0}_{J-1}\right] \text{ and } \left[h_{p,1}(\mathbf{x}_m|\mathbf{y}_\ell), \underbrace{0, \ldots, 0}_{J-1}\right]^T, \text{ respectively. The same}$$

formulation is adopted for each particle velocity component.

Then, we have

$$\mathbf{q} = \left[\mathbf{q}_1^T, \mathbf{q}_2^T, \ldots, \mathbf{q}_L^T\right]^T,$$

$$\mathbf{H}_p(\mathbf{x}_m) = \left[\mathbf{H}_p^T(\mathbf{x}_m|\mathbf{y}_1), \mathbf{H}_p^T(\mathbf{x}_m|\mathbf{y}_2), \ldots, \mathbf{H}_p^T(\mathbf{x}_m|\mathbf{y}_L)\right]^T,$$

$$\mathbf{H}_{v,c}(\mathbf{x}_m) = \left[\mathbf{H}_{v,c}^T(\mathbf{x}_m|\mathbf{y}_1), \mathbf{H}_{v,c}^T(\mathbf{x}_m|\mathbf{y}_2), \ldots, \mathbf{H}_{v,c}^T(\mathbf{x}_m|\mathbf{y}_L)\right]^T, c \in x, y, z,$$

which are of size $LJ \times 1$, $LJ \times (K + J - 1)$, and $LJ \times (K + J - 1)$, respectively.

In a similar way, the desired sound pressure and particle velocity at position $\mathbf{x}_m$ can be expressed as

$$p_n^d(\mathbf{x}_m) = s_n * g_{p,n}(\mathbf{x}_m) = \mathbf{g}_p^T(\mathbf{x}_m)\mathbf{s}_n \tag{12}$$

$$\mathbf{v}_n^d(\mathbf{x}_m) = s_n * g_{v,n}(\mathbf{x}_m) = \mathbf{g}_v^T(\mathbf{x}_m)\mathbf{s}_n \tag{13}$$

where $\mathbf{g}_p(\mathbf{x}_m)$ and $\mathbf{g}_v(\mathbf{x}_m)$ denote the RIR of pressure and particle velocity, respectively, from the virtual desired source to the $m$th control point.

While the conventional pressure-matching-based method aims to minimize the desired and reproduce sound pressure only, in this work, a joint control of sound pressure and particle velocity is investigated. That is, the mean squared error (MSE) between the desired and reproduced sound pressure and particle velocity are minimized simultaneously over $N$ time samples and $M$ control points. The cost function is formulated as follows:

$$J = \frac{1}{NM} \sum_{m=1}^{M} \sum_{n=1}^{N} (1-\tau)\Big(p_n(\mathbf{x}_m) - p_n^d(\mathbf{x}_m)\Big)^2 + \tau\|\mathbf{v}_n(\mathbf{x}_m) - \mathbf{v}_n^d(\mathbf{x}_m)\|_2 \tag{14}$$

where $p_n^d(\mathbf{x}_m)$, $p_n(\mathbf{x}_m)$, $\mathbf{v}_n^d(\mathbf{x}_m)$, and $\mathbf{v}_n(\mathbf{x}_m)$ denote the desired and reproduced sound pressure, the desired and the reproduced particle velocity at the control point $\mathbf{x}_m$, respectively. Note that in (14), the first term of sound pressure control is a scalar, while the second term of particle velocity control is a vector, which can be further represented as

$$||\mathbf{v}_n(\mathbf{x}_m) - \mathbf{v}_n^d(\mathbf{x}_m)||^2 =$$
$$\Big(v_n^x(\mathbf{x}_m) - v_n^{x,d}(\mathbf{x}_m)\Big)^2 + \Big(v_n^y(\mathbf{x}_m) - v_n^{y,d}(\mathbf{x}_m)\Big)^2 + \Big(v_n^z(\mathbf{x}_m) - v_n^{z,d}(\mathbf{x}_m)\Big)^2.$$

Equation (14) can also be represented in matrix form, i.e.,

$$\begin{aligned}
\mathbf{J}(\mathbf{q}) &= (1-\tau)(\mathbf{q}^T\mathbf{R}_p\mathbf{q} - 2\mathbf{q}^T\mathbf{r}_p + \sigma_p) + \tau(\mathbf{q}^T\mathbf{R}_v\mathbf{q} - 2\mathbf{q}^T\mathbf{r}_v + \sigma_v) \\
&= \mathbf{q}^T\mathbf{R}\mathbf{q} - 2\mathbf{q}^T\mathbf{r} + \sigma,
\end{aligned} \tag{15}$$

where $\tau$ denotes weighting parameter.

The spatial autocorrelation matrix is defined as

$$\mathbf{R} = (1-\tau)\mathbf{R}_p + \tau\mathbf{R}_v, \tag{16}$$

with $\mathbf{R}_p = \frac{1}{M}\sum_{m=1}^{M}\mathbf{H}_p(\mathbf{x}_m)\mathbf{R}_s\mathbf{H}_p^T(\mathbf{x}_m)$, $\mathbf{R}_s = \frac{1}{N}\sum_{n=1}^{N}\mathbf{s}_n\mathbf{s}_n^T$, and the same formulation for each particle velocity component, that is, $\mathbf{R}_c = \frac{1}{M}\sum_{m=1}^{M}\mathbf{H}_{v,c}(\mathbf{x}_m)\mathbf{R}_s\mathbf{H}_{v,c}^T(\mathbf{x}_m)$, $c \in x,y,z$, $\mathbf{R}_v = \mathbf{R}_x + \mathbf{R}_y + \mathbf{R}_z$.

The spatial cross-correlation vector is defined as

$$\mathbf{r} = (1-\tau)\mathbf{r}_p + \tau\mathbf{r}_v, \tag{17}$$

with $\mathbf{r_p} = \frac{1}{M}\sum_{m=1}^{M}\mathbf{H}_p(\mathbf{x}_m)\mathbf{R}_s\mathbf{g}_p^T(\mathbf{x}_m)$ and the same formulation for each particle velocity vector, $\mathbf{r_c} = \frac{1}{M}\sum_{m=1}^{M}\mathbf{H}_{v,c}(\mathbf{x}_m)\mathbf{R}_s\mathbf{g}_{v,c}^T(\mathbf{x}_m)$, $c \in x,y,z$, $\mathbf{r}_v = \mathbf{r}_x + \mathbf{r}_y + \mathbf{r}_z$.

The constant term is defined as

$$\sigma = (1-\tau)\sigma_p + \tau\sigma_v, \tag{18}$$

with $\sigma_p = \frac{1}{M}\sum_{m=1}^{M}\mathbf{g}_p^T(\mathbf{x}_m)\mathbf{R}_s\mathbf{g}_p(\mathbf{x}_m)$ and the same formulation for each particle velocity component, $\sigma_c = \frac{1}{M}\sum_{m=1}^{M}\mathbf{g}_{v,c}^T(\mathbf{x}_m)\mathbf{R}_s\mathbf{g}_{v,c}(\mathbf{x}_m)$, $c \in x,y,z$, $\sigma_v = \sigma_x + \sigma_y + \sigma_z$.

Minimizing the cost function Equation (14) by setting its derivative of $\mathbf{q}$ equal to 0, we get the solution

$$\hat{\mathbf{q}} = \mathbf{R}^{-1}\mathbf{r}. \tag{19}$$

### 3.2. EVD-Based Approach with Conjugate Gradient Algorithm

While the joint control of sound pressure and particle velocity for sound field reproduction is especially suited to non-uniform loudspeaker array setup with reduce number of loudspeakers and control points required, the proposed time-domain reproduction method has the potential to be used in real-time applications. However, for long-length RIRs and control filters, the inverse solution in (16) requires very high computational complexity. To solve this problem, the eigenvalue decomposition (EVD) based approach is adopted with the conjugate gradient (CG) method to search for the optimal solution in an iterative manner.

In (19), the matrix $\mathbf{R}$ is a symmetric positive definite matrix of size $LJ \times LJ$, where $L$ is the number of loudspeaker used for reproduction and $J$ is the length of the control filter for each loudspeaker. Solving the problem with the direct inverse operation requires $\mathcal{O}((LJ)^3)$ operations. Instead, we assume that the space spanned by the spatial autocorrelation matrix $\mathbf{R}$ can be approximated by its $I$ dominant eigen vectors, where $I \leq LJ$. Then, the CG method, which searches the solution in a set of orthogonal directions, can be used to find the solution iterative. The CG method adopted in this work has the advantage of reducing the computational complexity to $\mathcal{O}(I(LJ)^2)$ by setting the dimension of search direction as $I$. The flow of the algorithm is summarized in Table 1.

**Table 1.** Conjugate gradient algorithm for implementing the proposed time-domain sound field reproduction system.

---

**INITIALIZATION**:
1. Calculate the spatial autocorrelation matrix $\mathbf{R}$ and the spatial cross-correlation vector $\mathbf{r}$ using (16) and (17)
2. Set the initial value of the filter $\hat{\mathbf{q}}_1 = \mathbf{0}_{LJ \times 1}$, the initial search direction vector $\mathbf{d}_1 = \mathbf{r}$, and the initial residual error $\mathbf{e}_1 = \mathbf{r}$
3. Set the number of iterations $I$

**LOOP**: for $i = 1, 2, \ldots, I$

1. Determine the step $\alpha_i$ of the $i$th iteration according to $\alpha_i = \dfrac{\mathbf{e}_i^T \mathbf{e}_i}{\mathbf{d}_i^T \mathbf{R} \mathbf{d}_i}$
2. Update the estimates of the control filter $\hat{\mathbf{q}}_{i+1} = \hat{\mathbf{q}}_i + \alpha_i \mathbf{d}_i$ and the residual error $\mathbf{e}_{i+1} = \mathbf{e}_i - \alpha_i \mathbf{R} \mathbf{d}_i$
3. Calculate the factor $\beta_{i+1}$ that satisfies the conjugation condition, that is, $\beta_{i+1} = \dfrac{\mathbf{e}_{i+1}^T \mathbf{e}_{i+1}}{\mathbf{e}_i^T \mathbf{e}_i}$
4. Calculate the $i + 1$th search direction vector $\mathbf{d}_{i+1} = \mathbf{e}_{i+1} + \beta_{i+1} \mathbf{d}_i$

---

## 4. Simulation

In this section, we verify the effectiveness of the proposed reproduction method through numerical simulations in a room environment. For convenience, we treat the loudspeaker as a simple point source in this simulation. Two different loudspeaker layouts are simulated, i.e., a regular and an irregular loudspeaker array on the horizontal plane. Therefore, we consider both the desired sound field and reproduced sound field on the horizontal plane, for which only the two components of particle velocity along $x$ and $y$ axes are matched. In the simulation setup, the origin of the coordinate coincides with the left-bottom corner of the room and we use a segment of speech as the input signal of the system.

### 4.1. Performance Evaluation Metrics

Two performance evaluation metrics adopted are as follows:

- The normalized mean squared error (NMSE) of reproduced sound intensity, which is defined as

$$\epsilon = \frac{\frac{1}{NM} \sum_{m=1}^{M} \sum_{n=1}^{N} \|\mathbf{I}_n(\mathbf{x}_m) - \mathbf{I}_n^d(\mathbf{x}_m)\|^2}{\frac{1}{NM} \sum_{m=1}^{M} \sum_{n=1}^{N} \|\mathbf{I}_n^d(\mathbf{x}_m)\|^2} \tag{20}$$

with the sound intensity vector calculated as follows [23]:

$$\mathbf{I}_n(\mathbf{x}_m) = p_n(\mathbf{x}_m) \mathbf{v}_n(\mathbf{x}_m). \tag{21}$$

Here, $\mathbf{I}_n(\mathbf{x}_m)$ and $\mathbf{I}_n^d(\mathbf{x}_m)$ denote the reproduced and desired sound intensity at the point $\mathbf{x}_m$, respectively. The results over $N$ time samples and $M$ points are averaged in Equation (20).

Specially, the intensity reproduction NMSE along $c$ ($c \in x, y$) axis is investigated separately, that is,

$$\epsilon_c = \frac{\frac{1}{NM}\sum_{m=1}^{M}\sum_{n=1}^{N}\|\mathbf{I}_c(\mathbf{x}_m) - \mathbf{I}_c^d(\mathbf{x}_m)\|^2}{\frac{1}{NM}\sum_{m=1}^{M}\sum_{n=1}^{N}\|\mathbf{I}_c^d(\mathbf{x}_m)\|2} \tag{22}$$

Note that as proved in psycho-acoustic experiments, the sound intensity measure is closely linked with human perception of sound locations [24].

- The NMSE of the reproduced sound pressure, which defined as

$$\eta = \frac{\frac{1}{NM}\sum_{m=1}^{M}\sum_{n=1}^{N}\|p_n(\mathbf{x}_m) - p_n^d(\mathbf{x}_m)\|^2}{\frac{1}{NM}\sum_{m=1}^{M}\sum_{n=1}^{N}\|p_n^d(\mathbf{x}_m)\|^2}, \tag{23}$$

where $p_n(\mathbf{x}_m)$ and $p_n^d(\mathbf{x}_m)$ denote the reproduced and desired sound pressure at the point $\mathbf{x}_m$, respectively. This measure is commonly used for evaluating the accuracy of sound field reproduction systems.

### 4.2. Regular Loudspeaker Array

We first simulate the case of a regular circular loudspeaker array consisting of 8 evenly distributed loudspeakers, whose center is located at the center of the room and radius is 2 m . The dimension of the room is 8 m $\times$ 6 m $\times$ 4 m. Six control points (or matching microphones) form a concentric circle with the loudspeaker array, inside which is our sound reproduction zone. We have one matching point at the center of the circle, and the other five points evenly distributed on a circle with a radius of 0.2 m, as shown in Figure 1. The sampling frequency is set to 16 kHz. The RIRs are generated using the RIR Generator toolbox [25], which is based on the image source method [26]. The desired sound field comes from a point source located at $\mathbf{y}$ = (6 m, 5 m, 2 m).

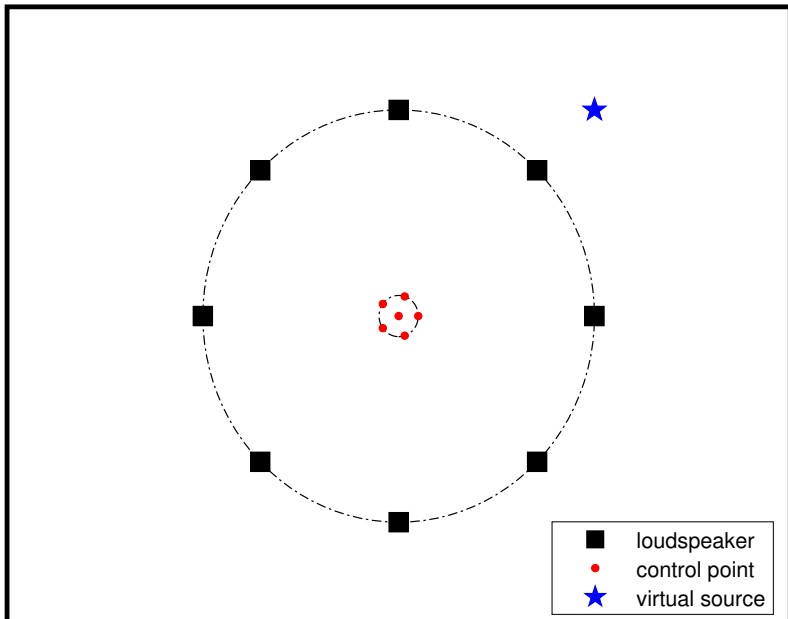

**Figure 1.** Simulation setup: 8 loudspeakers locate on a circle with a radius of 2 m, which are denoted by the black squares. The red dots denote the controlled points. The blue star indicates the location of the virtual sound source.

In Figure 2, we plot the NMSE of the reproduced intensity $\epsilon$ varying with the tuning parameter $\tau$. The case of $\tau = 0$ or $\tau = 1$ corresponds to controlling only the pressure or particle velocity, for which the reproduced sound intensity has a large error. The minimum intensity reproduction error occurs at $\tau = 0.5$, which is about 5 dB and 3 dB lower compared to the case of controlling only the pressure and the particle velocity, respectively. These results demonstrate that when the pressure and velocity are controlled with equal weights, which approximates sound-intensity control, the best reproduction performance can be obtained. As stated in literature, the sound intensity is closely related to source location perception [15], the proposed method also achieves the optimal reproduction results with sound intensity control under the regular loudspeaker array layout.

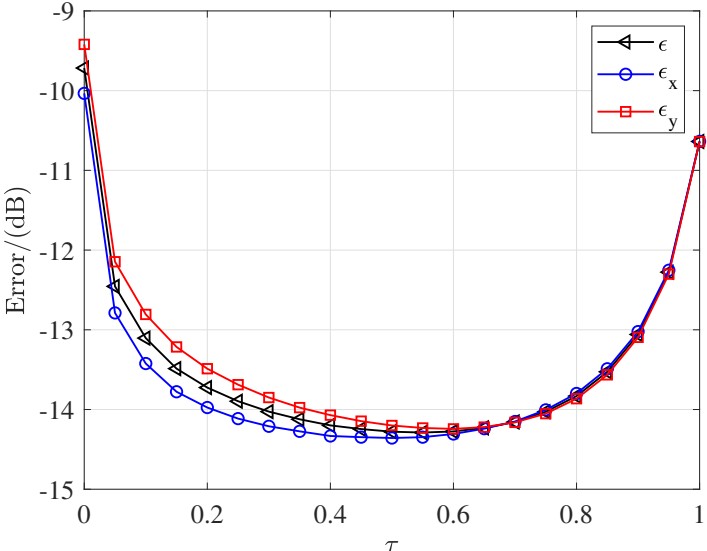

**Figure 2.** The NMSE of the reproduced intensity at control points with different tuning parameter $\tau$. The reverberation time $RT60$ = 200 ms and the control filter length $J = 400$.

We further investigate the validity of the proposed method in different reverberation conditions with varying control filter length. Figure 3 plots the NMSE of the reproduced sound intensity with the reverberation time $RT60$ increasing from 0.2 s to 0.7 s while the control filter length is set as a constant value of $J = 400$. As the reverberation time $RT60$ increase, the reproduction error also increases. In Figure 4, we change the control filter length, i.e., $J = 100, 200, 400, 800, 1000, 1600$, to examine the reproduction performance. When $J < 800$, the error decreases rapidly with the increasing control filter length; however, when $J$ is more than 800, the trend of the error decreases tends to be stable.

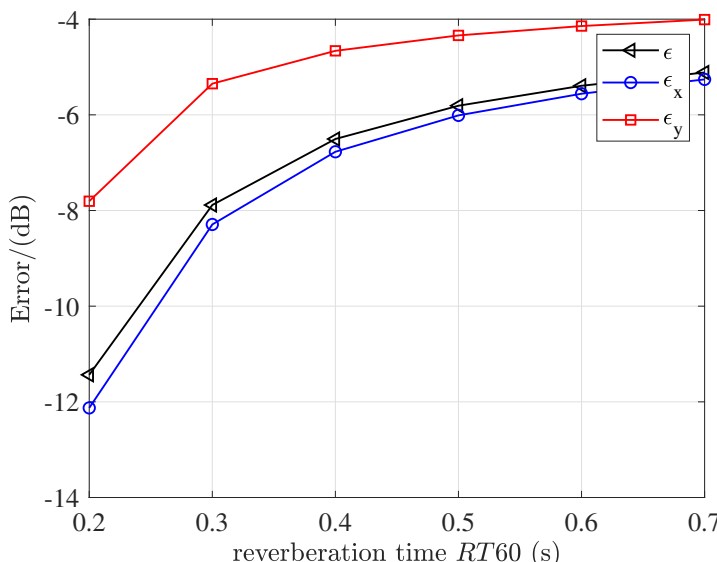

**Figure 3.** The NMSE of the reproduced intensity at control points under different reverberation times. The tuning parameter $\tau = 0.5$ and the control filter length $J = 400$.

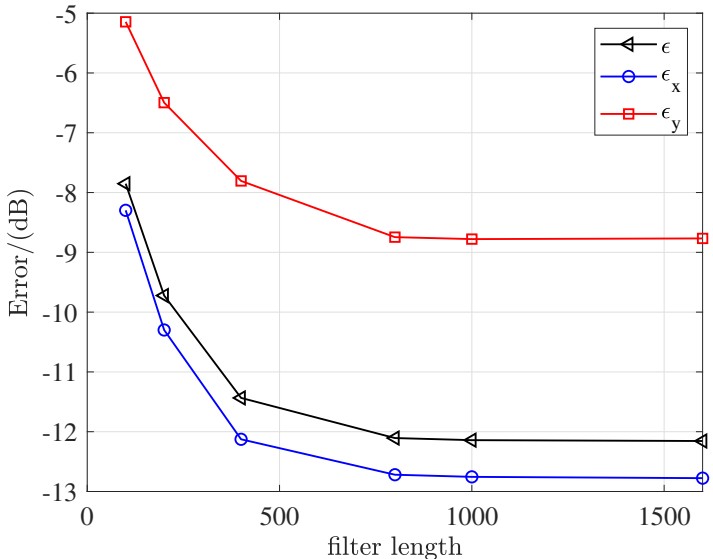

**Figure 4.** The NMSE of the reproduced intensity at controlled points with varying filter length. The tuning parameter $\tau = 0.5$ and reverberation time $RT60 = 200$ ms.

We then investigate the reproduction performance within the entire reproduction region, especially the reproduction results at uncontrolled points. We randomly selected 20 uncontrolled points, whose positions are shown in Table 2. The corresponding reproduction error $\epsilon$ and $\eta$ with different values of the tuning parameter $\tau$ are drawn in Figures 5 and 6, respectively. It can be seen that the NMSE value of both reproduced intensity and sound pressure firstly decreases and then increases at these uncontrolled points. The minimum error for the intensity reproduction and the sound pressure reproduction occurs at $\tau = 0.6$ and $\tau = 0.4$, respectively. Compared with previous works based on multi-point pressure matching which obtain the optimal reproduction performance only at the matching (or control) points, the above results demonstrate that the proposed method with an appropriate value of $\tau$ can achieve an accurate reproduction over an enlarged area, even within the entire control region. In other words, jointly controlling the sound pressure and particle velocity helps to enlarge the sound reproduction area.

**Table 2.** The 20 uncontrolled point positions.

| Uncontrolled Point No. | $x$ (m) | $y$ (m) | $z$ (m) |
|---|---|---|---|
| 1 | 4.0492 | 3.0087 | 2.0000 |
| 2 | 4.0070 | 3.0495 | 2.0000 |
| 3 | 3.9551 | 3.0219 | 2.0000 |
| 4 | 3.9653 | 2.9640 | 2.0000 |
| 5 | 4.0235 | 2.9559 | 2.0000 |
| 6 | 4.0693 | 3.0400 | 2.0000 |
| 7 | 3.9834 | 3.0783 | 2.0000 |
| 8 | 3.9204 | 3.0084 | 2.0000 |
| 9 | 3.9675 | 2.9269 | 2.0000 |
| 10 | 4.0595 | 2.9465 | 2.0000 |
| 11 | 4.0376 | 3.1034 | 2.0000 |
| 12 | 3.9133 | 3.0677 | 2.0000 |
| 13 | 3.9088 | 2.9385 | 2.0000 |
| 14 | 4.0303 | 2.8943 | 2.0000 |
| 15 | 4.1099 | 2.9962 | 2.0000 |
| 16 | 4.0900 | 3.1072 | 2.0000 |
| 17 | 3.9258 | 3.1187 | 2.0000 |
| 18 | 3.8642 | 2.9661 | 2.0000 |
| 19 | 3.9902 | 2.8603 | 2.0000 |
| 20 | 4.1298 | 2.9476 | 2.0000 |

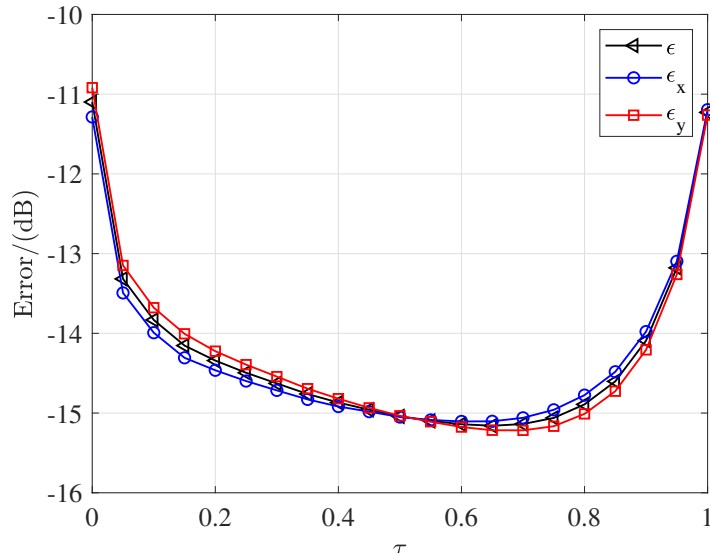

**Figure 5.** The NMSE of the reproduced intensity at uncontrolled points. The reverberation time $RT60 = 200$ ms and the control filter length $J = 400$.

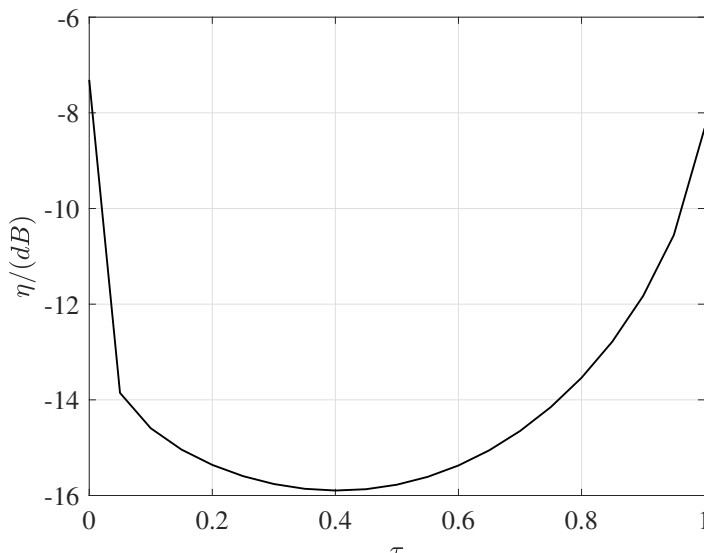

**Figure 6.** The NMSE of the reproduced sound pressure at uncontrolled points. The reverberation time $RT60 = 200$ ms and the control filter length $J = 400$.

### 4.3. Irregular Loudspeaker Array

Next, we investigate the proposed method on an irregular loudspeaker array. We adopt the widely used ITU-T standard 5.1 setup layout without the woofer unit to validate our method, as shown in Figure 7. In this setup, the angle between the left (right) and the center loudspeaker is $30°$, the angle between the surround left (surround right) and the center loudspeaker is $110°$, respectively. The other simulation setup is the same as Section 4.2.

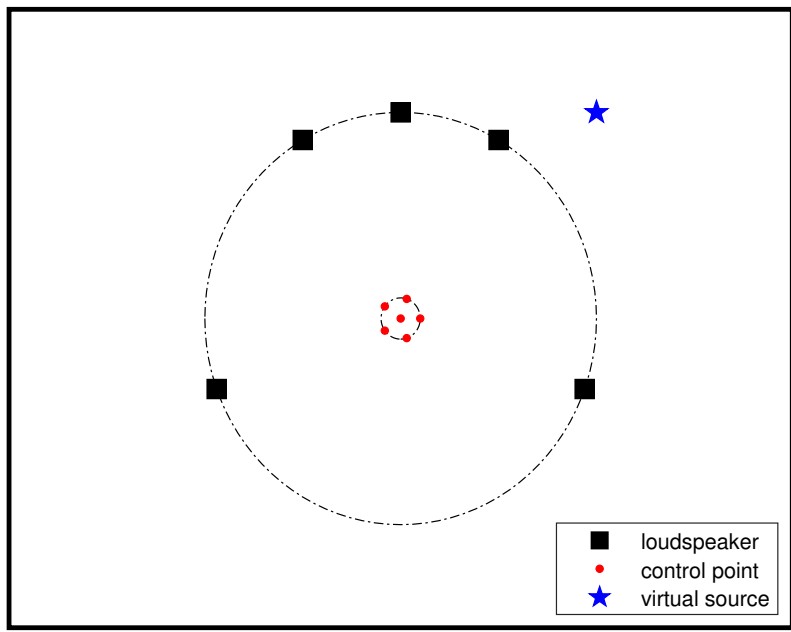

**Figure 7.** Simulation setup: The 5 loudspeakers are configured as the ITU-T standard 5.1, which are denoted by the black squares. The red dots denote the control points. The blue star indicates the location of the virtual sound source.

Figures 8 and 9 show the NMSE of the reproduced intensity $\epsilon$ varying with the tuning parameter $\tau$ at the control and the uncontrolled points, respectively. At both the controlled and uncontrolled points, the reproduction error $\epsilon$ shows a consistent trend, that is, first

decreasing and then increasing with the increase value of the tuning parameter $\tau$. The minimum error occurs when $\tau = 0.7$ in both Figures 8 and 9, proving that a jointly control of sound pressure and particle velocity with an adjustable weighting parameter has the flexibility to be adapted to the irregular loudspeaker array layout.

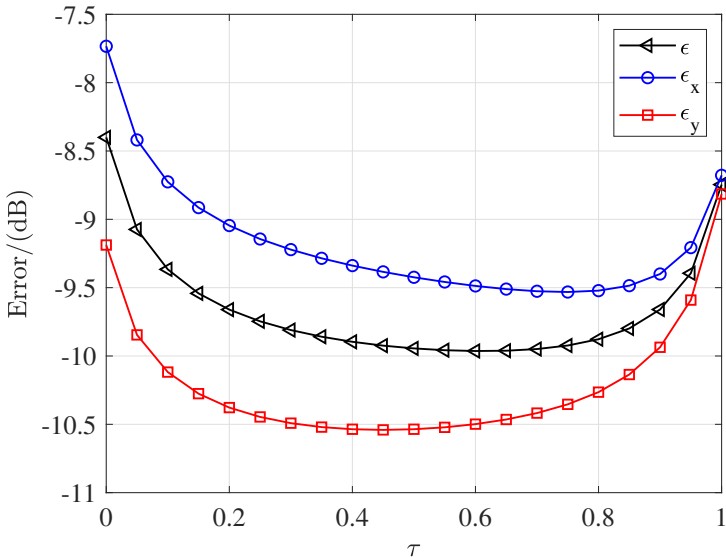

**Figure 8.** The NMSE of the reproduced intensity at controlled points. The reverberation time $RT60 = 200$ ms and the control filter length $J = 400$.

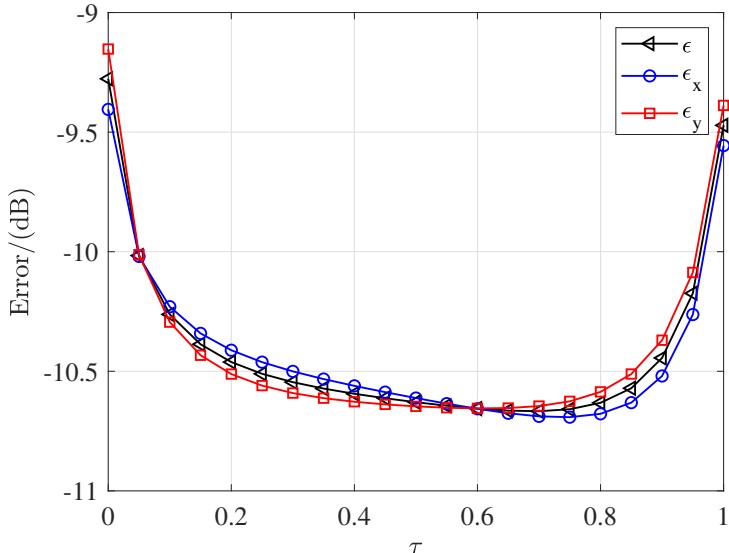

**Figure 9.** The NMSE of the reproduced intensity at uncontrolled points. The reverberation time $RT60 = 200$ ms and the control filter length $J = 400$.

Figures 10 and 11 show the NMSE of the reproduced sound pressure $\eta$ varying with the tuning parameter $\tau$ at the control and the uncontrolled points, respectively. Though Figure 10 indicates that the NMSE of the reproduced sound pressure monotonically increase as the parameter $\tau$ increase for reproduction at the control points. Plots in Figure 11, on the other hand, demonstrate that, at the uncontrolled points, the minimum sound pressure reproduction error occurs at $\tau = 0.2{\sim}0.4$, which is also lower than that of $\tau = 0$ or $\tau = 1$, i.e., the sole control of pressure and particle velocity. We can draw the conclusion that a joint control of sound pressure and particle velocity is beneficial to improve both sound pressure and sound intensity reproduction within the entire reproduction region.

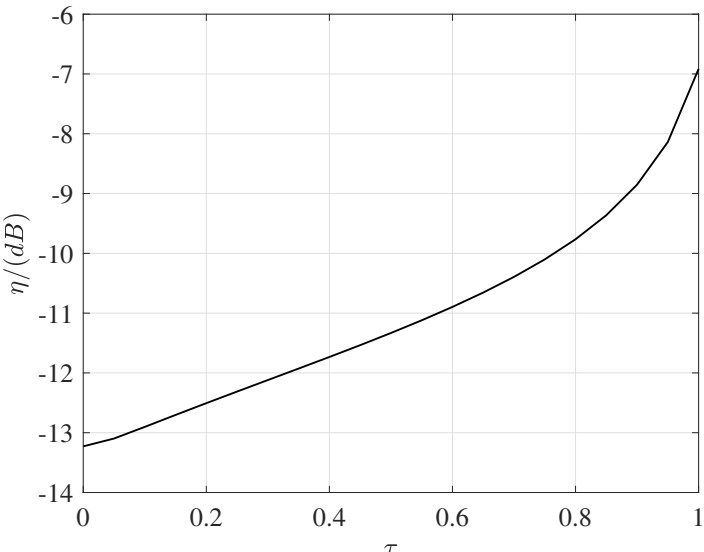

**Figure 10.** The NMSE of the sound pressure $\eta$ at controlled points. The reverberation time $RT60$ = 200 ms and the control filter length $J = 400$.

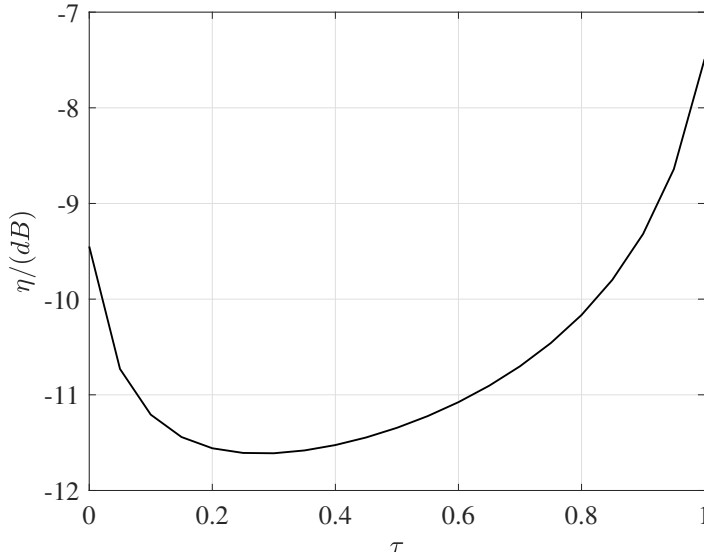

**Figure 11.** The NMSE of the sound pressure $\eta$ at uncontrolled points. The reverberation time $RT60$ = 200 ms and the control filter length $J = 400$.

*4.4. Computation Complexity Performance*

Finally, we examine the computational complexity performance of the proposed method, which uses the CG method to avoid a large-sized matrix inverse operation. We compared the processing time of the direct inverse operation and the CG method for implementing Equation (19). The run times were computed on a laptop with 2.4 GHz Intel(R) Core(TM) i5-1135G7 CPU with the algorithm simulated on the MATLAB R2020b. The cases of different iteration numbers, i.e., $I = 100, 200, 400, 800$, are simulated, in comparison with the direct inverse operation, and the results are shown in Table 3.

Figure 12 plots the NMES of the reproduced intensity $\epsilon$ using the direct inverse operation and the CG method with different iteration numbers, i.e., $I = 100, 200, 400, 800$. Combined with the results in Table 3, we can see that the CG method has almost the same reproduction accuracy as the direct inverse operation but significantly reduces the processing time.

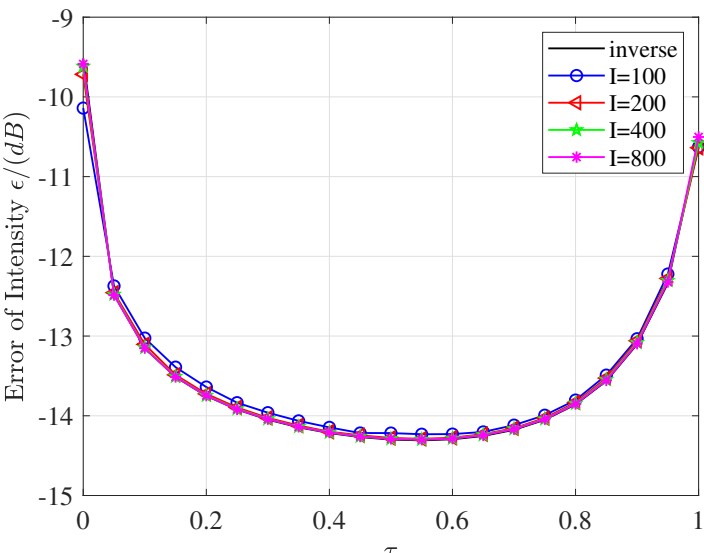

**Figure 12.** The NMSE of the reproduced intensity using the direct inverse and the adopted CG method. $I = 100, 200, 400, 800$ is the iteration number of CG. The reverberation time $RT60 = 200$ ms and the control filter length $J = 400$.

**Table 3.** The processing time of the direct inverse operation and the CG method, where the corresponding computational complexity is $\mathcal{O}((LJ)^3)$ and $\mathcal{O}(I(LJ)^2)$, respectively. $I$ is the iteration number of the CG method. The reverberation time $RT60 = 200$ ms and the control filter length $J = 400$.

| Method | Inverse | $I = 100$ | $I = 200$ | $I = 400$ | $I = 800$ |
|---|---|---|---|---|---|
| **Time**(s) | 203.97 | 25.53 | 29.90 | 39.02 | 70.75 |

## 5. Conclusions

We have proposed a time-domain sound field reproduction method with sound pressure and particle velocity jointly controlled. The control was formulated using a Lagrangian cost function with a tuning parameter to adjust the control weights, which gives the flexibility to achieve the optimal control at different loudspeaker array layouts. While most existing works implement particle velocity or sound intensity assisted sound field reproduction in frequency domain, the present work focused on time-domain reproduction and adopted the conjugate gradient method to reduce computational complexity. The proposed method was evaluated on both a regular loudspeaker array layout and an irregular loudspeaker array layout. We demonstrated that the proposed method improves both sound pressure and sound intensity reproduction with reduced computational complexity. Given that the reproduction system of controlling the particle velocity is especially suitable to a non-uniformly spaced loudspeaker array with reduced number of loudspeakers and control points required, the present work has the potential in real-time sound field reproduction applications when the reproduction environment is time varying, such as in-car audio systems.

**Author Contributions:** Conceptualization, W.Z. and X.H.; methodology, X.H.; validation, X.H. and J.W.; writing—original draft preparation, X.H. and J.W.; writing—review and editing, W.Z. and L.Z.; supervision, L.Z.; funding acquisition, W.Z. All authors have read and agreed to the published version of the manuscript.

**Funding:** This work was supported by the National Natural Science Foundation of China (NSFC) under Grant Nos. 61671380.

**Institutional Review Board Statement:** Not applicable.

**Informed Consent Statement:** Not applicable.

**Data Availability Statement:** Not applicable.

**Conflicts of Interest:** The authors declare no conflicts of interest.

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
