# Peer review of "Time-Domain Sound Field Reproduction with Pressure and Particle Velocity Jointly Controlled"

_applsci, doi:10.3390/app112210880_

Round 1

Reviewer 1 Report

Very good research work.

High level of scientific presentation.

Applicable methodology for highly qualified acoustic specialists and future software applications. 

More detailed information for the numerical results' verification will improve the quality of the paper. I recommend to authors to explain next steps of application of the presented methodology at the end.   

Reviewer 2 Report

The submitted paper is a good new approach to an important feature. I believe the paper deserves publication if minor aspects will be fixed. Most importantly, conclusions should me improved, mainly trying to discuss the present work with already known works in literature. Related to this, also introduction should explore a bit more the already published work and provide a better background to the work. In this regard, i find important to mention the work: Bianco, F., Teti, L., Licitra, G., & Cerchiai, M. (2017). Loudspeaker FEM modelling: Characterisation of critical aspects in acoustic impedance measure through electrical impedance. Applied Acoustics124, 20-29.
